# Novel Mitochondrial DNA Lineage Found among *Ochlerotatus communis* (De Geer, 1776) of the Nordic-Baltic Region

**DOI:** 10.3390/insects11060397

**Published:** 2020-06-26

**Authors:** Heli Kirik, Lea Tummeleht, Tobias Lilja, Olavi Kurina

**Affiliations:** 1Institute of Agricultural and Environmental Sciences, Estonian University of Life Sciences, Friedrich Reinhold Kreutzwaldi 5D, 51006 Tartu, Estonia; olavi.kurina@emu.ee; 2Institute of Veterinary Medicine and Animal Sciences, Estonian University of Life Sciences, Friedrich Reinhold Kreutzwaldi 62, 51006 Tartu, Estonia; lea.tummeleht@gmail.com; 3Department of Microbiology, Swedish National Veterinary Institute, 751 89 Uppsala, Sweden; tobias.lilja@sva.se

**Keywords:** *Ochlerotatus churchillensis*, *Ochlerotatus nevadensis*, *Ochlerotatus tahoensis*, barcoding, phylogenetics, speciation, vectors

## Abstract

The *Ochlerotatus (Oc.) communis* complex consist of three Northern American species as well as a common Holarctic mosquito (Diptera: Culicidae) *Oc. communis* (De Geer, 1776). These sister species exhibit important ecological differences and are capable of transmitting various pathogens, but cannot always be differentiated by morphological traits. To investigate the *Oc. communis* complex in Europe, we compared three molecular markers (COI, ND5 and ITS2) from 54 Estonian mosquitoes as well as two COI marker sequences from Sweden. These sequences were subjected to phylogenetic analysis and screened for *Wolbachia* Hertig and Wolbach symbionts. Within and between groups, distances were calculated for each marker to better understand the relationships among individuals. Results demonstrate that a group of samples, extracted from adult female mosquitoes matching the morphology of *Oc. communis*, show a marked difference from the main species when comparing the mitochondrial markers COI and ND5. However, there is no variance between the same specimens when considering the nuclear ITS2. We conclude that *Oc. communis* encompasses two distinct mitochondrial DNA lineages in the Nordic-Baltic region. Further research is needed to investigate the origin and extent of these genetic differences.

## 1. Introduction

*Ochlerotatus* (*Oc.*) *communis* complex includes four closely related mosquito species [1]: *Oc. communis* (De Geer, 1776), *Oc. churchillensis* (Ellis and Burst, 1973), *Oc. nevadensis* (Chapman and Barr, 1964) and *Oc. tahoensis* (Dyar, 1916). Morphology-based delimitation of these species is highly problematic due to a lack of reliable distinguishing traits, especially in adults [2]. Thus, researchers have employed both DNA sequencing, using mainly the mitochondrial cytochrome c oxidase subunit I (COI or COX1), and restriction fragment length polymorphism (RFLP) patterns to help with differentiation [3]. The namesake of the group, *Oc. communis*, is a common and often numerous Holarctic pest, whereas the other three species appear to be native to Northern America [1]. Due to the ubiquity of *Oc. communis* and because of its observed morphological variability, it is highly likely that this complex could have additional sister species in other parts of the world besides Northern America [2].

The phylogeography of the *Oc. communis* complex has received relatively little attention. At first these species were distinguished by morphologic as well as morphometric traits and the length differences of select loci, apparent in electrophoresis [1,4]. In 2014, the journal *Canadian Entomologist* published an article from H. H. Namin et al. describing barcoding (COI) results and designing a new diagnostic RFLP pattern for use with *Oc. communis*, *Oc. churchillensis* and *Oc. tahoensis* [3]. Since that, *Oc. churchillensis*, *Oc. nevadensis* and *Oc. tahoensis* have only been rarely sequenced, for example as part of vector disease investigations [5]. There are more studies on *Oc. communis*, but this species is still often diagnosed based on morphology alone, although genetic identification is also used [6,7,8,9,10]. Generally speaking, the *Oc. communis* complex does not currently appear to be under close study.

Mosquitoes from the *Oc. communis* complex have been associated with many pathogens. The Jamestown Canyon virus has been isolated from North American *Oc. communis* mosquitoes, which may be one of the species acting as an overwintering reservoir for the pathogen [5]. *Oc. communis* individuals, in some cases both adults and larva, have been found to carry Sindbis virus (known in Sweden as Ockelbo, in Finland as Pogosta and in Russia as Karelian virus) Batai virus, *Francisella (F.) tularensis* bacteria as well as different strains of the Inkoo virus in Scandinavian field studies [7,11,12]. *Oc. communis* could also be one of the main vectors of *Dirofilaria repens*, a filarial nematode which is currently expanding its area northward [8,13]. According to older studies, *Oc. communis* mosquitoes have also tested positive for Tahyna virus in Russia and six strains of California encephalitis virus in Canada [14,15].

Individuals within a phylogenetic group, even within isomorphic species, can often differ in their medical importance [16,17] and there is a noticeable lack of information regarding how the sister species within *Oc. communis* complex vary in their vector capacity and competence. Especially as some biological and ecological differences have been observed within the group. Firstly, *Oc. churchillensis* is the only autogenous species in the group and thought to be non-biting [1]. Both *Oc. nevadensis* and *Oc. tahoensis* seem to only be found in mountainous regions, the latter preferring higher elevations, while *Oc. churchillensis* inhabits forests near the North American tundra [4]. Because of these factors, it has been theorized that the sister species comprising *Oc. communis* complex may have derived from allopatric as well as sympatric speciation [3].

Maternally inherited *Wolbachia* Hertig and Wolbach, 1924 symbionts can also contribute to speciation within arthropods. *Wolbachia* is a genus of cytoplasmically transmitted bacteria that infect the tissues of many arthropods and some nematodes [18,19]. These endosymbionts have shown to cause cytoplasmic incompatibility, parthenogenesis and the death or feminization of biological males (reviewed by Correa and Ballard [20]). Because of this, *Wolbachia* infections have been seen as possible drivers of microevolution and even speciation [21,22,23]. *Wolbachia* strains have been detected in several different mosquito species, but infection rates vary [24,25]. At this time, no strains have been found in *Oc. communis* [8]. However, if detected, it could help explain some genetic results.

A larva with a COI sequence similar but not identical to *Oc. tahoensis* was recently found in Sweden [7], but the discovery was not further investigated. At the same time, similar cases were found with adult female mosquitoes in Estonia. Taking into account the possibility of additional *Oc. communis* complex species in Europe [2], a special attention was paid to the recently collected Estonian mosquitoes. The primary aim of this study was to search for a possible novel species within the *Oc. communis* complex. For this reason, both mitochondrial (mtDNA) and nuclear DNA (nDNA) markers from Nordic-Baltic mosquitoes were analyzed. These samples were also screened for infection with *Wolbachia* symbionts. Here we present the genetic information of 54 Estonian mosquitoes and compare those to two Swedish samples as well as to reference material from public nucleotide databases.

## 2. Materials and Methods

### 2.1. Sampling and Morphological Identification

This study is based on 54 Estonian mosquitoes and two Swedish COI sequences (Table A1). Of the Estonian individuals, 26 were morphologically identified as *Oc. communis*, six as *Oc. punctor* (Kirby, 1837), eight as *Oc. hexodontus* (Dyar, 1916), 13 as *Oc. cataphylla* (Dyar, 1916) and one as *Anopheles (An.) messeae* (Falleroni, 1926). While *Oc. communis* is the main focus, *Oc. punctor*, *Oc. hexodontus* and *Oc. cataphylla* samples were included in the analysis to compare how intra- as well as interspecific genetic variation of the *Oc. communis* complex relates to other common species of the genus *Ochlerotatus*. *An. messeae* was used as an outgroup. All Estonian mosquitoes were collected from six different sites during 2015–2016, using automated Mosquito Magnet^®^ Independence (Woodstream Corp., Lititz, PA, USA) machines. Four of these sampling sites were located on the Estonian mainland and two on the largest islands of the country—Saaremaa and Hiiumaa (Figure 1). Insects were stored at −20 °C until temporarily thawed and identified under a stereomicroscope Olympus SZ61 (Olympus Corporation, Shinjuku, Tokyo, Japan) to species level, using a standard taxonomic key [2]. Swedish COI sequences originate from Västerbotten County, first from a larva caught in 2014 [7] and the second from an adult female mosquito sent to the Swedish National Veterinary Institute (SVA) in 2017.

### 2.2. DNA Extraction

DNA was extracted from the Estonian mosquitoes using either DNeasy Blood and Tissue Kit (Qiagen, Hilden, North Rhine-Westphalia, Germany) or PrepMan^®^ Ultra Sample Preparation Reagent (Thermo Fisher Scientific Inc., Waltham, Massachusetts, USA). There were no qualitative differences between the used DNA extraction methods. Whole mosquitoes were used for the DNeasy Blood and Tissue Kit, while three legs from each specimen were taken for extracting DNA with the PrepMan^®^ Ultra Sample Preparation Reagent. DNeasy Blood and Tissue Kit was used in accordance with the manufacturers manual and the DNA extractions made with PrepMan^®^ Ultra Sample Preparation Reagent were conducted as specified in previous work [26].

### 2.3. DNA Markers

Two protein coding mitochondrial and two nuclear markers were amplified and sequenced from all Estonian mosquitoes used in this study. However, only three markers were used for further analysis as the D2 region of the large subunit 28S rDNA gene was too conserved between *Ochlerotatus* species to be of use, although it has been successfully utilized for species identification in other mosquito genera [27,28]. The 5’ region of the cytochrome c oxidase (COI) subunit I was chosen as one of the markers for its widespread use in mosquito identification and its generally good ability to differentiate between species, although it can at times either over- or underestimate the true number of well distinguished monophyletic groups [16,29,30]. The nicotinamide adenine dinucleotide (NADH) dehydrogenase subunit 5 (ND5) gene was used as an additional mitochondrial marker. ND5 is generally thought to have faster evolution rates compared to COI and thus has been used for inter- as well as intraspecies studies in mosquitoes [31,32,33]. Finally, the nuclear internal transcribed spacer 2 (ITS2), a region of the ribosomal RNA gene, was used as the most conserved marker. ITS2 sequences have been used for species identification in many animal groups and may be the most used nDNA marker for mosquitoes [16,34,35]. It has generally been proposed as a good marker to analyze alongside COI [16].

Each mosquito sample was also screened for *Wolbachia* symbionts by amplifying part of the *Wolbachia* surface protein (WSP) gene.

### 2.4. Primers

For COI, which is by far the most commonly sequenced marker for mosquitoes, we used the universal primers LCO1490 (5′-GGTCAACAAATCATAAAGATATTG G-3′) and HCO2198 (5′-TAAACTTCAGGGTGACCAAAAAATCA-3′), that consistently resulted in 710 bp long sequences [36]. A 450 bp segment of the ND5 region was amplified and sequenced using the primers 6500 (5′-TCCTTAGAATAAAATCCCGC-3′) and 7398 (5′-GTTTCTGCTTTAGTTCATTCTTC-3′) which were originally designed for *Aedes (Ae.) albopictus* [37]. Primer pair 5.8S (5′-TGTGAACTGCAGGACACATG-3′) and 28S (5′-ATGCTTAAATTTAGGGGGTA-3′) was used for ITS2, producing approximately 368 bp to 387 bp long sequences [38]. The ITS2 primers were initially developed to differentiate between cryptic Anopheline mosquitoes, but can be used for other mosquito genera as well. DNA from the *Wolbachia* symbionts WSP gene was amplified with the primers wsp 81F (5′-TGGTCCAATAAGTGATGAAGAAAC-3′) and wsp 691R (5′-AAAAATTAAACGCTACTCCA-3′) that would have resulted in 600 bp sequences [39].

### 2.5. Polymerase Chain Reaction (PCR) and Sequencing

While the composition of the polymerase chain reaction (PCR) mixes remained the same throughout, thermal cycler programs were adjusted for each primer pair to maximize the amplification yield of the respective marker regions. All PCR reactions contained 12.5 μL DreamTaq DNA Polymerase Master Mix (2X) (Thermo Fisher Scientific Inc., Waltham, MA, USA), 0.4 µM of each primer (0.04 µmol, TAG Copenhagen, Frederiksberg, Denmark), 10.5 μL nuclease-free water and 1 μL DNA template. The PCR program for amplifying the COI marker region was as follows: 95 °C for 2 min 15 s, followed by 35 cycles of 95 °C for 30 s, 57 °C for 45 s, 72 °C for 45 s and a final elongation step at 72 °C for 5 min. Although mostly identical, the PCR program for the ITS2 sequence introduced a much lower annealing temperature: 95 °C for 2 min 15 s, 35 cycles of 95 °C for 30 s, 45 °C for 45 s, 72 °C for 45 s and an elongation at 72 °C for 5 min. On the other hand, amplifying the mitochondrial ND5 region required a much longer PCR program, which contained 2 different sets of cycles on low annealing temperatures. The final program was: 94 °C for 3 min denaturation followed by 10 cycles of 94 °C for 30 s, 38 °C for 30 s, 65 °C for 45 s, then 50 cycles of 94 °C for 30 s, 38 °C for 30 s, 65 °C for 45 s and a last elongation of 65 °C for 3 min. The WSP region of the symbiont was amplified using a previously published PCR program [24] and a positive control sample from *Culex pipiens* was also added to the PCR. All samples were amplified with ESCO Swift Maxi Thermal Cycler (ESCO Micro Pte. Ltd., Changi South Street, Singapore, Singapore).

PCR products were checked for signals by electrophoresis using 1.6% agarose gel infused with ethidium bromide. Amplified samples were tinted with DNA Gel Loading Dye (6X) (Thermo Fisher Scientific Inc., Waltham, MA, USA) prior to electrophoresis. Positive signals were compared to GeneRuler 100 bp DNA Ladder, ready-to-use (Thermo Fisher Scientific Inc., Waltham, MA, USA) or, in the case of the WSP samples, to the GeneRuler 100 bp Plus DNA Ladder, ready-to-use (same company) to visually determine the approximate lengths of the replicated DNA strands. PCR products were cleaned and sequenced with Applied Biosystems 3130xl Genetic Analyzer by a two-directional procedure (Thermo Fisher Scientific Inc., Waltham, MA, USA).

### 2.6. Sequence Analysis 

Forward and reverse DNA strands were aligned and assembled into consensus sequences in BioEdit Sequence Alignment Editor version 7.2.5 [40]. Primers and low-quality areas were trimmed to produce the final sequences. MEGA X version 10.0.5 [41] was used for sequence alignment and data analysis. All original DNA sequences were uploaded to the online database GenBank. Reference sequences were added to the analysis via the Blast Search tool in MEGA X (Table A2). This was effective for COI and ITS2 markers. However, the mtDNA marker ND5 has so far received little attention in mosquitoes from the *Ochlerotatus* genus and, therefore, no previous sequences could be found from the database.

Protein-coding gene sequences were aligned based on codons, whereas the DNA strands for ITS2 were aligned by nucleotides alone and allowed gaps for indels. In all cases, the Multiple Sequence Comparison by Log-Expectation (MUSCEL) tool with default options was used for aligning sequences. The Find Best-Fit Substitution Model feature was used for all of the analyzed markers to determine the most appropriate model and Rates among Sites variable. These results were then used to calculate mean within and between group genetic distances measuring the proportion of nucleotide sites with differences between each sequence pair (p-distances). This was done using the Compute within Group Mean Distance and Compute between Group Mean Distance functions in MEGAX. Gaps/Missing Data Treatment was set to complete deletion, which ensured that all sequences of the same marker were trimmed to identical lengths: 441 bp for COI, 321 bp for ND5 and 251 bp for the ITS2 marker region.

Phylogenetic trees were constructed with the maximum likelihood method, while also using the analysis model and Rates among Sites recommended by the Find Best-Fit Substitution Model feature. The Gamma Parameter (+G) was set to 6, Gaps/Missing Data Treatment was set to complete deletion and Bootstrap with 1000 replications was employed each time. All trees were annotated and rooted using *An. messeae* as an outgroup. Trees were then modified to only display bootstrap values > 75% and distance values ≥ 0.01. Differences between population sizes were not accounted for. While *Oc. communis*, *Oc. punctor* and *Oc. cataphylla* are all common in Estonia, there is very little information about their exact effective population sizes. However, mosquitoes of the *Oc.* sp. group likely have a much smaller effective population size than the normal type *Oc. communis*.

## 3. Results

Of the 54 Estonian mosquitoes used in this study, 26 were identified as *Oc. communis* by morphological evaluation, but only 14 of these were grouped together by all three DNA markers. The remaining 12 individuals formed a separate monophyletic group within the *Oc. communis* complex based to their mitochondrial markers, similarly to two COI sequences received from Sweden. However, this pattern was not apparent when examining the nDNA results.

According to the phylogenetic tree based on the COI marker (Figure 2), 12 sequences from mosquitoes caught at three different sites (sites 1, 2 and 5) in Estonia (Figure 1) and 2 sequences from Sweden cluster together (hereafter referred to as *Oc.* sp.), distinct from *Oc. communis* and closer to the North American species *Oc. tahoensis* and *Oc. churchillensis*. In fact, there is on average about 0.063 (standard error (S.E) 0.018) substitutions per base difference between *Oc. communis* specimens and the *Oc.* sp. group. Meanwhile, the p-distances between *Oc.* sp. individuals and *Oc. tahoensis* as well as *Oc. churchillensis* are smaller, 0.046 (S.E. 0.013) and 0.054 (S.E. 0.015) respectively. It should also be noted, that the *Oc.* sp. COI sequences show a high similarity to each other, being almost genetically identical. This is in contrast to the larger genetic diversity within other groups. These results show that there is a previously unknown group of genetically distinct individuals belonging to the *Oc. communis* complex found in Europe.

The ND5 marker region also suggests that mosquitoes forming the *Oc.* sp. group are a distinct genetic unit, separate from traditional *Oc. communis* mtDNA sequences (Figure 3). Compared to the COI region, the ND5 marker sequences are even more variable between groups. Between *Oc. communis* and the *Oc.* sp. cluster, there is a difference of on average 0.083 (S.E 0.031) base substitutions per site. Yet, within group average evolutionary distances remain small. In the *Oc.* sp. group, there are only on average 0.001 (S.E 0.001) base substitutions per site over all of the sequence. The p-distance is once again larger among the traditional *Oc. communis* samples, averaging 0.008 (S.E 0.009) base substitutions. ND5 marker sequences associated with *Oc. punctor*, *Oc. hexodontus* and *Oc. cataphylla* clusters are even more variable. Unfortunately, there are no ND5 marker sequences from *Oc. tahoensis*, *Oc. churchillensis* or *Oc. nevadensis* currently available in GenBank. Likewise, because the ND5 region is a less popular marker than COI and ITS2, there were also no reference sequences for *Oc. communis*, *Oc. punctor*, *Oc. hexodontus* or *Oc. cataphylla*. However, the ND5 sequences included in this study support the conclusions drawn from the COI marker.

While mitochondrial markers outlined *Oc.* sp. group as a separate entity, this is not the case for the nuclear marker ITS2. In fact, there are no differences between the ITS2 sequences from *Oc. communis* and the *Oc.* sp. samples (Figure 4). On the other hand, *Oc. churchillensis* reference sequences downloaded from GenBank maintain their genetic distance from *Oc. communis*, being on average 0.025 (S.E. 0.009) substitutions per base apart. The same is true for *Oc. punctor*, *Oc. hexodontus* and *Oc. cataphylla* clusters, which differ on average from the *Oc. communis* group by 0.035 (S.E 0.011) and 0.079 (S.E 0.017) base substitutions per site, respectively. It should also be noted that the ITS2 sequences of the *Oc. communis* cluster have no notable within-group genetic variance. Within group average genetic variance is also relatively low in other groups. From these results we can see that the ITS2 region of Estonian *Oc. communis* individuals is quite conserved and does not echo the variance shown by the mtDNA markers.

Estonian mosquito samples were screened for *Wolbachia* surface protein as it could offer an explanation for the divergent mtDNA lineage within the analyzed *Oc. communis* individuals. However, there were no positive electrophoresis signals for any of the samples except for the positive control. Therefore, there is no evidence of *Wolbachia* symbionts in the *Oc. communis* individuals used in this study.

## 4. Discussion

There is a discrepancy between the mtDNA and nDNA markers among mosquitoes morphologically identified as *Oc. communis* in the Nordic-Baltic region. The mitochondrial markers COI and ND5 clearly distinguish two clusters of individuals within the *Oc. communis* complex: One identical to *Oc. communis* reference sequences and the other representing a new group. However, this difference is not reflected in the ITS2 sequences of the same mosquitoes. Based on the nuclear marker, all examined *Oc. communis*-like individuals have identical ITS2 marker sequences, whereas reference sequences for *Oc. churchillensis*, which also belongs to the *Oc. communis* complex, remain distinct from *Oc. communis*. There appears to be a previously unreported mtDNA lineage within the Nordic-Baltic *Oc. communis* populations, but no variance within the nDNA to point to a distinct species.

Both COI and ND5 marker regions have high support for similar monophyletic clades and both trees show some amount of nucleotide changes between traditional *Oc. communis* individuals, while *Oc.* sp. sequences are largely identical. This shows, that the *Oc.* sp. clade is evolutionarily younger than the traditional *Oc. communis* and its members probably less numerous, as the mtDNA of this group has yet to accrue many mutations [44]. However, the nuclear marker ITS2 shows noticeably less divergence between all of the analyzed *Ochlerotatus* species. Moreover, the D2 region of the large subunit 28S, also originally sequenced for this study, was unable to provide any notable differences within *Ochlerotatus*, although it has been shown to work for anopheline mosquitoes [28]. It seems that the species analyzed for this study are in general more closely related to each other than those most often sequenced in other mosquito genera. Therefore, it is currently difficult to say if any of the markers used in this work might underestimate the true diversity of the *Ochlerotatus* species. This matter would be greatly improved by finding and analyzing nDNA markers with better resolution for the *Ochlerotatus* genus. It would also be useful to sequence a larger number of *Oc. communis*, in order to obtain a better overview of its natural intraspecific variance.

The reason for mitochondrial differences between coexisting groups can be hard to pinpoint. As the common type *Oc. communis* and the individuals with differing mtDNA coexist in the same communities and have identical ITS2 sequences, it is probable that they are intermixing. MtDNA variation in arthropod populations can be influenced by symbiotic bacteria like the maternally inherited *Wolbachia* [45]. However, none of the samples used in this study were positive for *Wolbachia* DNA and this coincides with previous observations [8]. There could be other speciation-driving factors in play, but that is impossible to determine at this juncture. Importantly, the fauna of the Nordic-Baltic region is relatively young, only emerging after the Last Glacial Period more than 9000 years ago when it was recolonized by organisms from different glacial refugiums [46,47,48]. It is possible that Eurasian *Oc. communis* complex individuals with the differing mtDNA used to be geographically separated, but not genetically different enough as to have evolved reproductive barriers.

Uncertainties in mtDNA sequencing results can also be caused by pseudogenes or heteroplasmy. Many organisms are known to have nuclear insertions of mitochondrial sequences (NuMts) as well as multiple types of functional mtDNA in the same individual (heteroplasmy)—both of these can interfere with the amplification and sequencing of mitochondrial markers, resulting in erroneous conclusions if not recognized [49]. Sequences from COI and ND5 genes have been known to be incorporated into nDNA as NuMts [50]. Furthermore, starting with *Aedes aegypti* (Linnaeus, 1762) and *Culex quinquefasciatus* (Say, 1823), NuMts have now been found in many mosquito species [51,52]. However, these insertions in nDNA are non-functional and, therefore, not under the same constraints or mutation rates as the real mtDNA [53,54]. These pseudogenes tend to contain inappropriate stop-codons and indels as well as more point mutation than normal [49,55]. Such problems are not evident in the sequences analyzed in this study. Heteroplasmy can also result in ambiguous reads in Sanger sequencing, but can be harder to identify compared to NuMts [16]. It cannot be completely ruled out that the less common *Oc. communis* mtDNA lineage reported in this study is a sign of heteroplasmy. 

While *Oc. communis* is not currently considered an important disease vector, it has been indicated as a possible carrier of several viral and bacterial pathogens. This species is known for its wide distribution and can be numerous at times [2]. Because of this, genetic deviations within *Oc. communis* populations may be important from future vector and pest control standpoints. Estonian and Swedish faunas are relatively young and it is reasonable to assume that *Oc. communis* individuals carrying the different mtDNA variant are not limited to the Nordic-Baltic region, but can be found within a much wider area. However, there has thus far been little indication of notable genetic discrepancies within this species in Central and Western Europe. Because of this, more sampling efforts should be directed towards Eastern Eurasia. Also, there is more work to be done in regards to sequencing additional nDNA markers from *Oc. communis* complex species and finding more informative nDNA markers to use with genus *Ochlerotatus* in general.

## 5. Conclusions

The current study presents evidence for an additional discrete mtDNA lineage within *Oc. communis* in Europe. This common Holarctic pest is the namesake of the *Oc. communis* complex, which it shares with three closely related Northern American species. While these sister species have sometimes been regarded as subspecies to *Oc. communis*, there are clear differences in their ecology, genetic material and in some cases morphology. It has been long theorized that there could be more species belonging to the *Oc. communis* complex in other parts of the world besides Northern America. In this study we show, based on the COI and ND5 markers, that there is a group of *Oc.* sp. individuals with a distinct mtDNA lineage within morphologically identified *Oc. communis* mosquitoes in Estonia and Sweden. We also show that these differences are not apparent in the nDNA of the same individuals. It was also determined that the analyzed mosquitoes had no detectable *Wolbachia* infection, ruling these maternally inherited symbionts out as a possible explanation for the mitochondrial differences.

## Figures and Tables

**Figure 1 insects-11-00397-f001:**
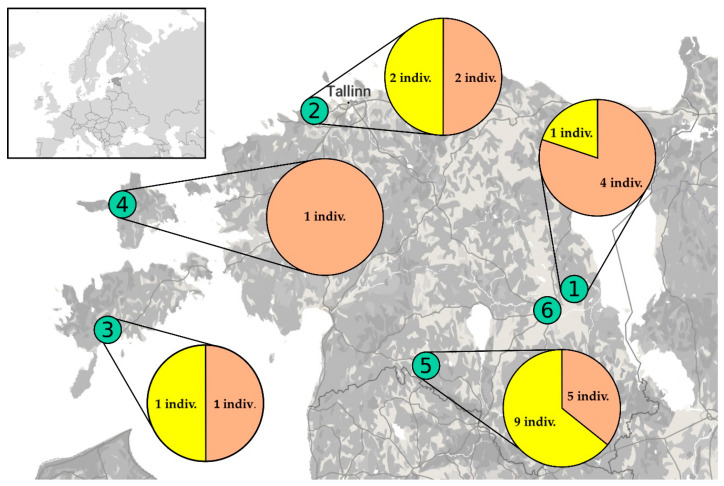
Map showing the six Estonian collection sites, indicated with green circles. Pie charts demonstrate the amount of *Oc.* sp. (yellow background) and *Oc. communis* (orange background) specimen caught from each site. In total this paper includes 20 mosquitoes from site 1 “Undi” (58°29′ N, 26°54′ E), eight from site 2 “Muraste” (59°28′ N, 24°27′ E), four from site 3 “Mändjala” (58°13′ N, 22°20′ E), five from site 4 “Vanajõe” (58°53′ N, 22°26′ E), 16 specimens from site 5 “Metsaküla” and one mosquito from site 6 “Tartu” (58°04′ N, 25°31′ E). 31 of these mosquitoes were caught in 2015 and 23 during 2016. Base map curtesy of ©OpenStreetMap contributors (https://www.openstreetmap.org/copyright) and ©MapTiler (https://www.maptiler.com/copyright/).

**Figure 2 insects-11-00397-f002:**
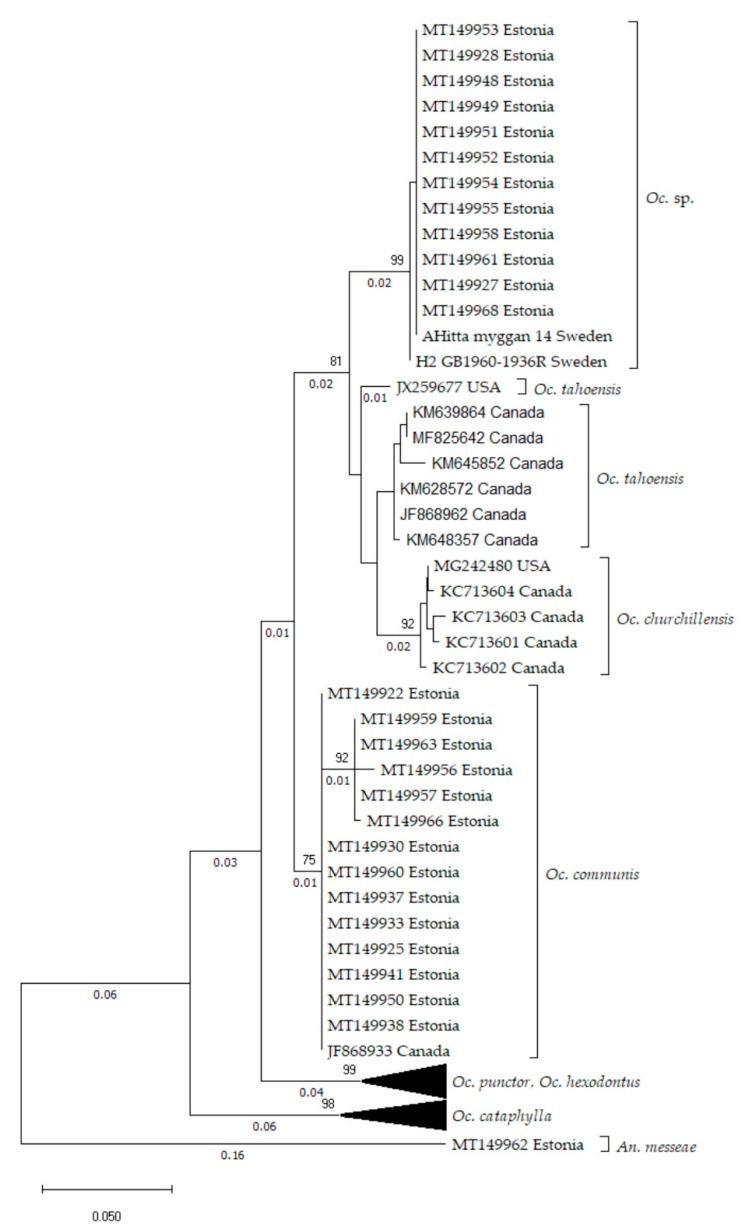
Phylogenetic tree based on 72 partial cytochrome c oxidase (COI) sequences (441). Calculated with the Maximum Likelihood method, using Tamura 3-parameter [42] model, with discrete gamma distribution (6 categories (+G, parameter = 0.1790)) Branch lengths are shown to scale and measured as the number of substitutions per site. *An. messeae* was used as an outgroup for rooting the tree. *Oc.* sp. sequences are genetically closer to North American species *Oc. tahoensis* and *Oc. churchillensis* than the widespread *Oc. communis*.

**Figure 3 insects-11-00397-f003:**
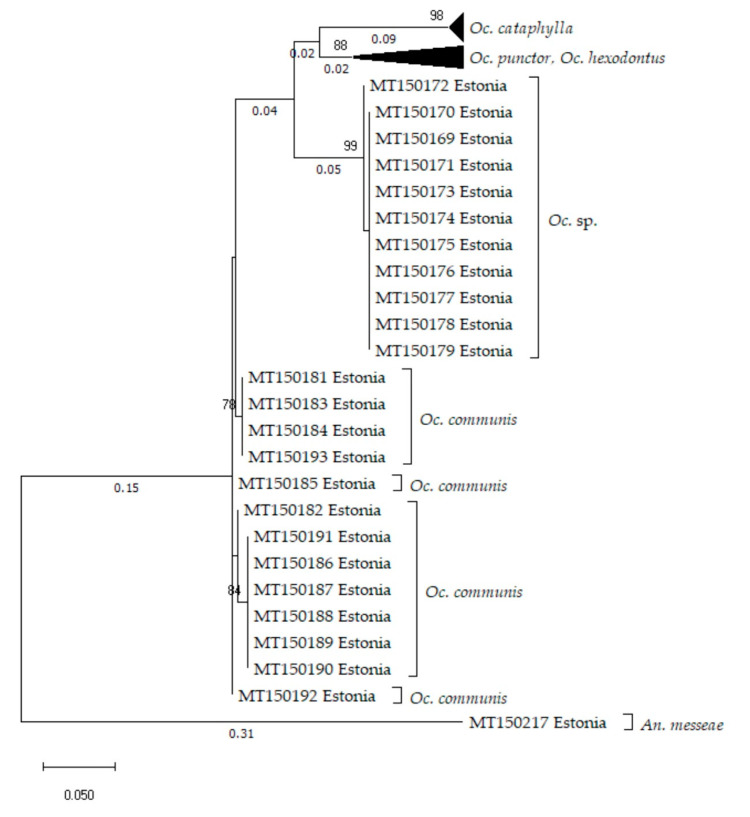
Phylogenetic tree representing the information of 48 dehydrogenase subunit 5 (ND5) marker region sequences (321 bp). The tree is calculated with the maximum likelihood method, using the Tamura 3-parameter [42] model and a discrete gamma distribution (6 categories (+G, parameter = 0.1516)). Branch lengths are shown to scale and measured based on the average number of substitutions per site between sequence pairs. *An. messeae* was used the outgroup in order to root the tree. *Oc.* sp. sequences cluster together, away from the *Oc. communis* group.

**Figure 4 insects-11-00397-f004:**
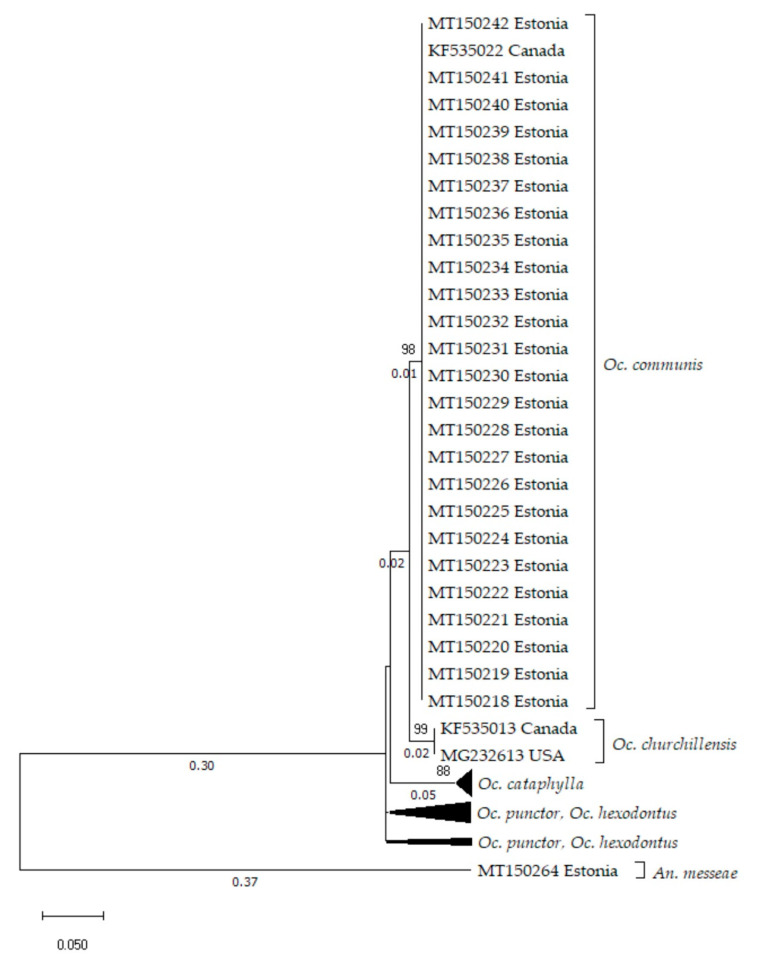
Phylogenetic tree based on the 53 internal transcribed spacer 2 (ITS2) marker sequences (251 bp). Calculated with the maximum likelihood method, using the Kimura 2-parameter model [43] and discrete Gamma distribution (6 categories (+G, parameter = 0.8737)). Branch lengths are shown to scale and measured based on the average number of substitutions per site between sequence pairs. *An*. *messeae* was used as the outgroup for rooting the tree. There appears to be no variation amid the ITS2 sequences *Oc.* sp. sequences and *Oc. communis*. However, *Oc. churchillensis* sequences obtained from GenBank remain as a separate group.

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
