# Peer review of "Novel Mitochondrial DNA Lineage Found among Ochlerotatus communis (De Geer, 1776) of the Nordic-Baltic Region"

_insects, 2020, doi:10.3390/insects11060397_

Round 1

Reviewer 1 Report

This paper analyses the Ochlerotatus communis species complex, using samples from Estonia and Sweden. Using two mtDNA markers and one nuclear marker, the authors report two mtDNA lineages in Estonia, but only a single nuclear DNA lineage.

The paper is well-written and clear. I think that the results will be informative for researchers studying these taxa, though I do think a second (or third, etc) nuclear marker would help strengthen their results and assist greatly with interpreting the results.

I have several points to raise about the paper. Primarily, if there is no nuclear divergence, how does it follow that there might be multiple species here, as suggested throughout the paper? The lack of nuclear divergence should be evidence of interbreeding, no? Or is it that the nuclear marker is not sufficiently powerful or too unconserved to detect differences? Either is possible. It could be that the two mtDNA lineages split in the past and diverged, but then after some paleohistorical event they joined back together.

Following the above, I recommend a  more careful consideration of how each marker is thought to operate. With only 3 markers, it’s important to understand differences between them. I would also strongly recommend including the mtDNA lineages on a map, to show the reader if they are found in distinct areas or mixed together. This would provide assistance with interpreting the results.

Introduction

This Introduction is well-written and the descriptions are clear.

There needs to be more attention given in the Introduction to what genetic work has already been done on these and/or related species. There also needs to be more attention to describing what the purpose of the study is.

L81: ‘morphology’ should read ‘morphologically’

It would be good to introduce here or in the Methods how each marker is expected to ‘behave’ – is the marker conserved or rapidly evolving?

Methods

The Methods are detailed as the laboratory protocols. However, more information is needed on analysis here. You mention genetic distances in the Results but I see no mention of this in the Methods. What sort of genetic distances are these? How did you calculate them? Did you account for differences in population size between species (which biases genetic distance calculations)?

I’m assuming distances are calculated from the phylogenetic analysis but this needs to be explicit here.

Results

L180 See above. What do these genetic distances refer to?

Figure 4 It seems that there is little genetic variation generally for this nuclear marker, even across space and between different taxa. Is ITS2 highly conserved, and is this thus expected?

Discussion

It seems the result of low differentiation for the nuclear marker, which is a major part of the paper, is difficult to interpret. It would be much clearer if there were deeper divergences among taxa or across space, but this does not seem to be the case.

I think that for some of the points in this Discussion there would need to be a clearer signal in the nuclear DNA to make the claim – is there another marker you can use that is more highly variable? If not, some of these assertions will have to be toned down or removed.

There is much in this Discussion that doesn’t relate clearly to the study, while there’s much to talk about in the study not mentioned here. Please keep the Discussion focused on the results. This includes an honest appraisal of the limitations of this study e.g. using 3 markers rather than high-throughput sequencing as predominantly used these days.

Author Response

I have several points to raise about the paper. Primarily, if there is no nuclear divergence, how does it follow that there might be multiple species here, as suggested throughout the paper? The lack of nuclear divergence should be evidence of interbreeding, no? Or is it that the nuclear marker is not sufficiently powerful or too unconserved to detect differences? Either is possible. It could be that the two mtDNA lineages split in the past and diverged, but then after some paleohistorical event they joined back together.

Thank you pointing this out. Yes, no nuclear divergens would point to interbreeding. Generally ITS2 has been considered to be a very informative marker for mosquitoes alongside COI. However, by far most of the genetic work has been done on Anopheles, which are malaria vectors, and some Aedes species, which can be very invasive. Although mosquitoes in the genus Ochlerotatus (Oc.) also contribute to the world’s disease burden, they have seen much less interest. Because of this, there are less scientific work to compare our results to and it is more difficult to evaluate the ITS2 results of our study. It is also interesting, that when we sequenced 28S for this work, which has also been successfully used in differentiating between Anopheles species, it was too conserved to be helpful in our case. I think that it could be indeed a case where two slightly different groups joined back together at some point.

Following the above, I recommend a more careful consideration of how each marker is thought to operate. With only 3 markers, it’s important to understand differences between them.

Thank you for this succession, we have added discussion of the used markers to both the Materials and Methods section as well as the discussion. It is usually thought that out of the three markers, ND5 is the fastest evolving and can also be used for population studies. COI and ITS2 are believed to have good resolution in mosquito studies. ITS2 is the most conserved in this case.

I would also strongly recommend including the mtDNA lineages on a map, to show the reader if they are found in distinct areas or mixed together. This would provide assistance with interpreting the results.

Thank you for the recommendation. We have added graphics to the map showing the amount of Oc. sp and Oc. communis caught from different sites. One collection site had neither.

Introduction

This Introduction is well-written and the descriptions are clear.

There needs to be more attention given in the Introduction to what genetic work has already been done on these and/or related species. There also needs to be more attention to describing what the purpose of the study is.

Thank you, we have updated the Introduction and added information about previous studies involving the Oc. communis complex. We have also reworked the ending part of the Introduction to make the aim clearer.

L81: ‘morphology’ should read ‘morphologically’

Thank you for pointing this out, it is now fixed.

It would be good to introduce here or in the Methods how each marker is expected to ‘behave’ – is the marker conserved or rapidly evolving?

Thank you for the comment. We have added an introduction to each marker to the Methods section. Hopefully this makes the Results and Discussion easier to understand.

Methods

The Methods are detailed as the laboratory protocols. However, more information is needed on analysis here. You mention genetic distances in the Results but I see no mention of this in the Methods. What sort of genetic distances are these? How did you calculate them? Did you account for differences in population size between species (which biases genetic distance calculations)?

I’m assuming distances are calculated from the phylogenetic analysis but this needs to be explicit here.

Thank you for pointing this out. We have now updated the Methods section to better reflect the analysis. Genetic distances were calculated as the mean proportion of nucleotide sites with changes between sequence pairs (also known as p-distance). These were calculated using the appropriate functions in MEGAX and corrected for backward and parallel nucleotide substitutions using the models recommended by the same program.

We accounted for the variation in sequence lengths by letting MEGAX automatically trim sequences, but unfortunately we could not account for the differences in population sizes. Oc. communis, Oc. punctor, Oc. hexodontus and Oc. cataphylla are all very common and often numerous in Estonia. Sadly there are no studies about the effective population sizes of these mosquitoes in this country. However, it is of course clear, that the population of the individuals with the differing mtDNA must be much smaller than the population of the Oc. communis with the normal mtDNA type.

Results

L180 See above. What do these genetic distances refer to?

Thank you for bringing this to our attention. We have added to the Methods section that the genetic distance talked about in our study is p-distance.

Figure 4 It seems that there is little genetic variation generally for this nuclear marker, even across space and between different taxa. Is ITS2 highly conserved, and is this thus expected?

This result was not expected. ITS2 is the most widely used nDNA marker for mosquitoes and has worked well for species identification in many genera. However, it seems to be less suitable for the species used in this study. The same goes for partial 28S, which was far to conserved to distinguish between the Ochlerotatus species, although it is quite useful for Anopheles. We have added this discussion to the article.

Discussion

It seems the result of low differentiation for the nuclear marker, which is a major part of the paper, is difficult to interpret. It would be much clearer if there were deeper divergences among taxa or across space, but this does not seem to be the case.

Thank you, we have the same uncertainty. This is exasperated by the fact, that there are very few sequences from the other Oc. communis complex species available in GenBank and that genus Ochlerotatus is studied less often than Anopheles.

I think that for some of the points in this Discussion there would need to be a clearer signal in the nuclear DNA to make the claim – is there another marker you can use that is more highly variable? If not, some of these assertions will have to be toned down or removed.

Thank you for alerting us to this issue. It is indeed true, that the ITS2 results are unfortunately slightly more difficult to interpret in this study. We have toned down the Discussion section and tried to include a more balanced explanation of the results.

There is much in this Discussion that doesn’t relate clearly to the study, while there’s much to talk about in the study not mentioned here. Please keep the Discussion focused on the results. This includes an honest appraisal of the limitations of this study e.g. using 3 markers rather than high-throughput sequencing as predominantly used these days.

Thank you, we have rewritten some of the discussion and removed text that seemed to veer too far from the issue at hand. Indeed, high-throughput sequencing would make working with large numbers of insects easier and finding Oc. communis individuals with differing mtDNA faster. However, we would have still targeted the same marker regions, as those are the ones with reference material in GenBank. Also, while many high-throughput machines can sequence many DNA strands at the same time, most technologies sequence short fragments which is less useful for DNA barcoding purposes.

Additionally, sequencing whole mosquito genomes is sadly not yet cost-effective for these types of studies. Approaches using techniques to sequence reduced genomes such as ddRAD sequencing can lower the cost but is still expensive and demands both specialized molecular biology equipment and bioinformatic handling beyond the scope of this study.

Reviewer 2 Report

In this study, the authors have used several molecular markers to characterize wild-caught Ochlerotatus communis mosquitoes collected in Estonia. These results indicate that there is some level of differentiation within these mosquitoes, particularly based on the mitochondrial markers. This group is important because it contains species that are both nuisance mosquitoes and vectors of disease, making them relevant to public health. This type of study also helps us to understand the dynamics of evolution and adaptation in mosquitoes, particularly those that are abundant and have a wide geographic range. It is also valuable to have reference sequences for these different loci as a contribution from this study to aid research into these species. This manuscript is extremely well written and clear.

Major Comments:

Lines 178- 181 - For the per base substitution calculations mentioned here, the average can be skewed if the sequences being compared are not evenly sized. Did the authors trim the sequences to the same length before calculating the base substitution average? This should be more explicit in the methods section and the bp length of analysed sequences made clear for this calculation for the discussion of each loci.

Figure 2 – The species used as the outgroup should be noted in the figure, not just in the text. This should be applied to the other tree figures as well.

Minor Comments:

Phylogeny figures - suggest making “Oc.” in “Oc. sp.” italic to match the other text in the figure, as it is still a genus name and should be italicized.

Lines 45-51 – Several Ochlerotatus species have been shown to be capable of being infected with West Nile Virus, I think it is worth mentioning in this section as an example of how this group of mosquitoes might potentially contribute to the spread of emerging viruses.

I don’t know what the normal standards of this journal are, but perhaps the methods section could be broken up into distinct sections such as “PCR reactions” and “Phylogenetic analysis”, whatever is most consistent with the journal style.

The ITS2 sequences being less differentiated than the other markers is interesting, it may indicate that this is an emerging cryptic group of these mosquitoes and not yet a separate species since the CO1 loci is faster evolving than ITS2. The relative evolutionary rates of these different loci may be worth mentioning in the text.

Lines 228-232 - Since there were no Wolbachia in the collected specimens, I don’t know if it is really worth mentioning. If this paragraph remains in the text, suggest mentioning in a short line in the abstract that no specimens were found positive.

Line 316-317 – Since you have a separate Conflict of Interest section immediately after this line, the statement here about no conflict of interest is not needed.

Finally, ensure that genus and species names in the References are properly italicised.

Author Response

Major Comments:

Point 1: Lines 178- 181 - For the per base substitution calculations mentioned here, the average can be skewed if the sequences being compared are not evenly sized. Did the authors trim the sequences to the same length before calculating the base substitution average? This should be more explicit in the methods section and the bp length of analysed sequences made clear for this calculation for the discussion of each loci.

Response 1: Thank you for pointing this oversight out. Sequences were not all trimmed to the same length, instead we used the option to disregard all bases with caps/deletions in MEGAX when calculating distances and phylogenetic trees. This gives virtually the same result and we have updated the article to make this approach clearer. To be certain, I also ran the genetic distance and phylogenetic analyses with manually trimmed COI sequences and had the same result.

Point 2: Figure 2 – The species used as the outgroup should be noted in the figure, not just in the text. This should be applied to the other tree figures as well.

Response 2: Thank you for this remark. We have made the appropriate changes to the figures and to the text accompanying them.

Minor Comments:

Point 3: Phylogeny figures - suggest making “Oc.” in “Oc. sp.” italic to match the other text in the figure, as it is still a genus name and should be italicized.

Response 3: Thank you once again, the figures have been updated. I apologize for this oversight.

Point 4: Lines 45-51 – Several Ochlerotatus species have been shown to be capable of being infected with West Nile Virus, I think it is worth mentioning in this section as an example of how this group of mosquitoes might potentially contribute to the spread of emerging viruses.

Response 4: Thank you for the suggestion. Indeed, some Ochlerotatus (Oc.) species have been brought up in connection to the West Nile Virus (WNV), especially Oc. caspius. However, none of the Oc. communis complex species have thus far been indicated to my knowledge. Also, Oc. communis is not known to frequently bite birds. Because of this we have hesitated in including the WNV in the list of possible pathogens in this article. Of course, Oc. communis complex individuals are not always tested for WNV, so this information could change with time.

Point 5: I don’t know what the normal standards of this journal are, but perhaps the methods section could be broken up into distinct sections such as “PCR reactions” and “Phylogenetic analysis”, whatever is most consistent with the journal style.

Response 5: Thank you for pointing this out. It seems that the articles published in Insects do indeed use subheadings in their methods section. This certainly makes the text easier to understand and we have made the necessary changes

Point 6: The ITS2 sequences being less differentiated than the other markers is interesting, it may indicate that this is an emerging cryptic group of these mosquitoes and not yet a separate species since the CO1 loci is faster evolving than ITS2. The relative evolutionary rates of these different loci may be worth mentioning in the text.

Response 6: Thank you, we have also found this curious, as ITS2 is widely used for differentiating between mosquito species. It is commonly accepted, that ND5 is generally faster evolving than COI and ITS2 is the slowest evolving of the three. We actually also sequenced the D2 region of 28S for this work, but found that it was too conserved in Ochlerotatus, even though it has been successfully used in Anopheles. It seems, that the species analyzed in our study are over-all genetically quite closely related. We have added this explanation to the article.

Point 7: Lines 228-232 - Since there were no Wolbachia in the collected specimens, I don’t know if it is really worth mentioning. If this paragraph remains in the text, suggest mentioning in a short line in the abstract that no specimens were found positive.

Response 7: Thank you for the suggestion, we have cut the text concerning Wolbachia shorter in the discussion. Hopefully we have been able to make the text more cohesive.

Point 8: Line 316-317 – Since you have a separate Conflict of Interest section immediately after this line, the statement here about no conflict of interest is not needed.

Response 8: Thank you. We removed the indicated sentence.

Point 9: Finally, ensure that genus and species names in the References are properly italicised.

Response 9: Thank you for the reminder. We fixed the genus and species names and excessive capital letters in the references.

Round 2

Reviewer 1 Report

Well done making these changes. The manuscript looks good in its present form

Author Response

Dear Reviewer,

Thank you once again for the previous corrections and thank you for the vote of confidence. I am glad you approve of the changes.

Sincerely yours,

Heli Kirik